# The Cruciality of Single Amino Acid Replacement for the Spectral Tuning of Biliverdin-Binding Cyanobacteriochromes

**DOI:** 10.3390/ijms21176278

**Published:** 2020-08-30

**Authors:** Keiji Fushimi, Hiroki Hoshino, Naeko Shinozaki-Narikawa, Yuto Kuwasaki, Keita Miyake, Takahiro Nakajima, Moritoshi Sato, Fumi Kano, Rei Narikawa

**Affiliations:** 1Graduate School of Integrated Science and Technology, Shizuoka University, 836 Ohya, Suruga, Shizuoka 422-8529, Japan; fushimi.keiji@shizuoka.ac.jp (K.F.); hoshino.hiroki.15@shizuoka.ac.jp (H.H.); miyake.keita.17@shizuoka.ac.jp (K.M.); 2Core Research for Evolutional Science and Technology, Japan Science and Technology Agency, 4-1-8 Honcho, Kawaguchi, Saitama 332-0012, Japan; cmsato@mail.ecc.u-tokyo.ac.jp; 3Cell Biology Center, Institute of Innovative Research, Tokyo Institute of Technology, Nagatsuta, Midori-ku, Yokohama, Kanagawa 226-8503, Japan; narikawa.n.ab@m.titech.ac.jp (N.S.-N.); kano.f.aa@m.titech.ac.jp (F.K.); 4Graduate School of Arts and Sciences, University of Tokyo, 3-8-1 Komaba, Meguro, Tokyo 153-8902, Japan; kuwasakiyuto@g.ecc.u-tokyo.ac.jp (Y.K.); ctnaka@g.ecc.u-tokyo.ac.jp (T.N.); 5Kanagawa Institute of Industrial Science and Technology, 705-1 Shimoimaizumi, Ebina, Kanagawa 243-0435, Japan; 6Research Institute of Green Science and Technology, Shizuoka University, 836 Ohya, Suruga, Shizuoka 422-8529, Japan

**Keywords:** spectral diversity, spectral tuning, mammalian intrinsic chromophore, near-infrared, optogenetics, fluorescence imaging, dark reversion

## Abstract

Cyanobacteriochromes (CBCRs), which are known as linear tetrapyrrole-binding photoreceptors, to date can only be detected from cyanobacteria. They can perceive light only in a small unit, which is categorized into various lineages in correlation with their spectral and structural characteristics. Recently, we have succeeded in identifying specific molecules, which can incorporate mammalian intrinsic biliverdin (BV), from the expanded red/green (XRG) CBCR lineage and in converting BV-rejective molecules into BV-acceptable ones with the elucidation of the structural basis. Among the BV-acceptable molecules, AM1_1870g3_BV4 shows a spectral red-shift in comparison with other molecules, while NpF2164g5_BV4 does not show photoconversion but stably shows a near-infrared (NIR) fluorescence. In this study, we found that AM1_1870g3_BV4 had a specific Tyr residue near the d-ring of the chromophore, while others had a highly conserved Leu residue. The replacement of this Tyr residue with Leu in AM1_1870g3_BV4 resulted in a blue-shift of absorption peak. In contrast, reverse replacement in NpF2164g5_BV4 resulted in a red-shift of absorption and fluorescence peaks, which applies to fluorescence bio-imaging in mammalian cells. Notably, the same Tyr/Leu-dependent color-tuning is also observed for the CBCRs belonging to the other lineage, which indicates common molecular mechanisms.

## 1. Introduction

Cyanobacteriochromes (CBCRs), which are known as linear tetrapyrrole-binding photoreceptors, to date can only be detected from cyanobacteria [1,2]. Only a cGMP-phosphodiesterase/adenylate cyclase/FhlA (GAF) domain is required for chromophore ligation and proper photoconversion, while two or three domains, including the GAF domain, are required for those in the case of the distantly related phytochromes. The CBCR GAF domains are categorized into several lineages, and the expanded red/green (XRG) CBCR lineage is one of the most characterized [3,4,5,6,7,8,9,10,11,12,13,14,15,16,17,18,19]. The typical XRG CBCR GAF domains covalently ligate phycocyanobilin (PCB) and show reversible photoconversion between the red-absorbing Pr form (15*Z*–PCB in dark states) and the green-absorbing Pg form (15*E*–PCB in photoproduct states) [3,4,5]. Three-dimensional structures of both forms have provided the structural basis for the mechanism of photoconversion of this lineage [20,21,22]. 

We have recently revealed that the XRG CBCR GAF domains from the chlorophyll *d*-bearing cyanobacterium *Acaryochloris marina* covalently incorporate biliverdin (BV, Figure 1A) and show reversible photoconversion between the far-red-absorbing form (Pfr) (15*Z*–BV in the dark state) and the orange-absorbing form (Po) (15*E*–BV in the photoproduct state) [23,24]. Based on the comparison between the BV-acceptable and the BV-rejective molecules, we have identified four residues crucial for BV incorporation (BV4) [25]. In some cases, the introduction of BV4 into the BV-rejective molecules resulted in efficient BV incorporation. We have further revealed the crystal structure of one of such molecules in its Pfr form [25]. One of the engineered BV-binding CBCR GAF domains, AM1_1870g3_BV4, has shown red-shifted spectral property for the Pfr form in comparison with the other BV-binding molecules. However, the molecular mechanism behind this unique red-shifting event has not been elucidated.

The BV-binding molecules have attracted much attention for application to optogenetics and fluorescence bio-imaging mainly for two reasons [26,27,28,29]. First, because BV is a chromophore intrinsic to mammalian cells, heterologous expression of only the apoprotein would be sufficient for these applications. Second, far-red light absorbed by BV can penetrate into the deep mammalian tissues. These properties are advantageous for in vivo optogenetic control and fluorescence bio-imaging. In fact, we have succeeded in visualizing the mouse liver without invasion by using the BV-binding near-infrared (NIR) fluorescent probe, NpF2164g5_BV4, which is another engineered BV-binding CBCR GAF domain that does not show photoconversion but stably shows NIR fluorescence [25]. Because longer wavelength light would penetrate into deeper mammalian tissues, rational red-shifting mutagenesis should greatly contribute to develop better NIR fluorescent probes for in vivo imaging.

In this study, we found that highly conserved Leu residue near the d-ring of the chromophore is replaced with Tyr specifically in AM1_1870g3_BV4 (Tyr_605_). The replacement of the Tyr residue with Leu resulted in a blue-shift of the Pfr form. Conversely, the replacement of the Leu residue with Tyr in NpF2164g5_BV4 (Leu_962_) resulted in a large red-shift of not only the absorption peak but also the fluorescence excitation and emission peaks, which applies to fluorescence bio-imaging in mammalian cells.

## 2. Results

### 2.1. Rational Site-Directed Mutagenesis for Spectral Tuning

Among the BV-binding XRG CBCR GAF domains, AM1_1870g3_BV4 showed red-shifted absorption spectra among the BV-binding CBCR molecules [25]. Thus, it was considered that AM1_1870g3_BV4 possessed specific residue(s) crucial for this red-shifted property. To identify such residue(s) crucial for spectral tuning, we compared the AM1_1870g3_BV4 sequence with those of the other BV-binding ones based on the structural information of the BV-binding AnPixJg2_BV4 (Figure 1B,C) [25]. As a result of the in silico analysis, Tyr_605_ was identified as a residue within 6 Å of the chromophore specific to AM1_1870g3_BV4, whereas the other molecules possessed Leu residue at this position. Because this Tyr/Leu position was located near the d-ring (Figure 1B), these residues may affect the π-conjugated system of the chromophore contributing to spectral tuning.

To verify this assumption, Tyr_605_ was replaced with Leu to generate AM1_1870g3_BV4_Y_605_L. His-tagged AM1_1870g3_BV4_Y_605_L was purified by nickel affinity chromatography (Figure 2). The purified AM1_1870g3_BV4_Y_605_L covalently incorporated BV and showed photoconversion and dark reversion (details of the dark reversion are described below) (Figure 2 and Figure 3A,B lower panel, and Table 1). An absorption peak of the dark state, Pfr form, was at 704 nm, which was 9 nm blue-shifted in comparison with that of the background molecule AM1_1870g3_BV4 peaking at 713 nm (Figure 3A upper panel, and Table 1). The photoproduct state of AM1_1870g3_BV4_Y_605_L showed broad absorbance from the orange to red region probably because of incomplete photoconversion, and we could not assign its peak wavelength (Figure 3A lower panel), which was similar to the case for AM1_1870g3_BV4 (Figure 3A upper panel). The incomplete photoconversion observed for these molecules may be due to relatively high dark reversion kinetics and low photoconversion quantum yield, which prevented us from determining the absolute absorption peaks of the photoproduct states of these molecules. Next, to obtain spectral information about the photoproduct state, we compared the Pfr-minus-photoproduct difference spectra of these two molecules (AM1_1870g3_BV4_Y_605_L and AM1_1870g3_BV4) and found that positive peaks corresponding to the dark states (Pfr forms) were at 708 and 718 nm, and negative peaks corresponding to the photoproduct states were at 605 and 619 nm, respectively (Figure 3C and Table 1). Not only the positive peak but also the negative peak of AM1_1870g3_BV4_Y_605_L were blue-shifted in comparison with those of AM1_1870g3_BV4, indicating that this replacement may also affect the photoproduct color-tuning. The SAR (Specific absorbance ratio; Pfr peak absorbance/Protein peak absorbance at 280 nm) values were calculated as 0.79 and 0.85, respectively, indicating BV-binding efficiency and molecular coefficient were comparable to each other (Table 1). In conclusion, these results were clearly consistent with our assumption, which states that the Tyr/Leu position is crucial for spectral tuning.

Because we succeeded in obtaining the loss-of-function molecule to absorb a shorter wavelength region, we next focused on the gain-of-function molecule to absorb a longer wavelength region. To transfer this basic insight into applied science, we focused on the BV-binding molecule, NpF2164g5_BV4, which has been proven to be applicable as a NIR-fluorescent probe without photoconversion, although shorter wavelength absorption peaking at 680 nm should be improved [25]. We replaced Leu_962_ with the Tyr residue to generate NpF2164g5_BV4_L_962_Y. His-tagged NpF2164g5_BV4_L_962_Y was purified by nickel affinity chromatography (Figure 2). The purified NpF2164g5_BV4_L_962_Y covalently incorporated BV and showed absorption around the far-red region (Pfr form, λ_max_ 697 nm) (Figure 2 and Figure 4A,B lower panel, and Table 1). Expectedly, the absorption peak of NpF2164g5_BV4_L_962_Y was largely red-shifted by 17 nm in comparison with the background molecule, NpF2164g5_BV4 (Pfr form, λ_max_ 680 nm) whose spectral shapes were comparable to each other (Figure 4A,B upper panel, and Table 1). Although we measured the excitation and emission fluorescence spectra of NpF2164g5_BV4_L_962_Y and NpF2164g5_BV4 at room temperature (r.t.), these spectra were highly broad (Figure 4C left). To characterize fluorescence property in detail, we further measured low-temperature fluorescence spectra under −196°C in liquid nitrogen. The excitation and emission fluorescence spectra of NpF2164g5_BV4_L_962_Y were largely improved, peaking at 711 nm and 728 nm, respectively, which also red-shifted in comparison with those of NpF2164g5_BV4 peaking at 696 nm and 707 nm, respectively (Figure 4C right and Table 1). The fluorescence quantum yield of NpF2164g5_BV4_L_962_Y was calculated as 2% and was lower than 4% of the background molecule NpF2164g5_BV4 (Table 1) [25]. The SAR values of the variant and the background molecule were calculated as 0.86 and 0.89, respectively, indicating BV-binding efficiency and molecular coefficient are comparable to each other (Table 1).

### 2.2. Dark Reversion

Because initial preliminary experiments clarified that both photoproduct states of AM1_1870g3_BV4 and AM1_1870g3_BV4_Y_605_L showed rapid dark reversion to the dark states (Pfr forms), we measured the Po absorption spectra at 10 °C by irradiating far-red light to repress the dark reversion (Figure 3A). Moreover, we calculated the half-lives of the dark reversion of these two molecules at 25 °C (Figure 3D and Table 1). The dark reversion half-life of the AM1_1870g3_BV4 was 7.3 s ± 0.03 at 25 °C, whereas that of BV4_Y_605_L was 20.6 s ± 0.11 under the same conditions. These molecules exhibited a moderately fast dark reversion in comparison with the other BV-binding molecules except for AnPixJg4_BV4 (2.8 s ± 0.10 at 25 °C), which showed 3- and 7-fold faster dark reversion than AM1_1870g3_BV4 and BV4_Y_605_L, respectively [25].

### 2.3. Fluorescence Imaging in Living Mammalian Cells Using NpF2164g5_BV4_L_962_Y

To demonstrate the applicability of NpF2164g5_BV4_L_962_Y for in vivo imaging, we tested whether the NIR fluorescence from NpF2164g5_BV4_L_962_Y could be observed in a living mammalian cell. HeLa cells were transfected with a humanized infrared fluorescent protein (iRFP), NpF2164g5_BV4, or NpF2164g5_BV4_L_962_Y that were N-terminally fused with an enhanced green fluorescent protein (EGFP). After incubation with BV, the cells were imaged using a confocal fluorescence microscope. The iRFP was a previously developed NIR-fluorescent probe incorporating BV and was used as a positive control in this study [26]. Although HeLa cells expressing only the EGFP tag emitted only green fluorescence without NIR fluorescence (Figure 5, EGFP), HeLa cells expressing the fused ones emitted not only green fluorescence but also NIR fluorescence (Figure 5, EGFP-hiRFP, EGFP-hNpF2154g5_BV4, and EGFP-hNpF2164g5_BV4_L_962_Y). Notably, the exciting red laser powers were 2%, 4%, and 15% for EGFP-hiRFP, EGFP-hNpF2154g5_BV4, and EGFP-hNpF2164g5_BV4_L_962_Y, respectively, indicating that the NIR fluorescence yield of EGFP-hNpF2164g5_BV4_L_962_Y was weaker than those of the other two molecules.

## 3. Discussion

In this study, we identified the cruciality of single amino acid replacement for spectral tuning of the XRG CBCRs. The replacement of Tyr with Leu in AM1_1870g3_BV4 resulted in a spectral blue-shift (Figure 3 and Table 1), whereas the converse replacement of Leu with Tyr in NpF2164g5_BV4 resulted in a spectral red-shift (Figure 4 and Table 1). The Leu residue of the homologous molecule, AnPixJg2_BV4 faces the chromophore d-ring (Figure 1B). Based on this structural arrangement, we can assume that the well-established trapped twist model would also be involved in this spectral tuning [1,16,30,31]. The d-ring of the dark state is twisted against the rings B–C plane in the AnPixJg2_BV4 structure, which has the Leu residue [20]. The bulky Tyr residue instead of the Leu residue may push the d-ring to facilitate a less twisted geometry, which would result in extended π-conjugated system. Instead, a phenolic hydroxy group in the Tyr residue may interact with the d-ring via a water molecule(s), which affects the localization of the π-electrons. These interactions would result in the spectral red-shift. In the case of the AM1_1870g3_BV4, not only the dark state but also the photoproduct state was suggested to be also red-shifted in comparison with the AM1_1870g3_BV4_Y_605_L variant based on the difference spectra (Figure 3 and Table 1). Because the Leu residue is also positioned near the d-ring in the photoproduct states of the homologous molecules, NpR6012g4 and Slr1393g3, the spectral tuning of the photoproduct state can also be explained in the context of the same trapped twist model [21,22]. We also found that the Y_605_L replacement in AM1_1870g3_BV4 resulted in modulation of the photoproduct stability, in which the Y_605_L variant showed a rather slower dark reversion than the background molecule (Figure 3C and Table 1). The Tyr/Leu residue is suggested to affect the d-ring geometry of the photoproduct state, which would contribute to the stability of the photoproduct. Notably, AM1_1870g3 possesses the Tyr residue at this position among the XRG CBCR lineage (Figure 1C), indicating that AM1_1870g3 would specifically acquire the Tyr residue for spectral tuning. 

Recently, we have reported the design of multicolored photo-switches based on a single CBCR scaffold, AM1_1499g1, which belongs to a certain DXCF lineage (lineage possessing highly conserved Asp-Xaa-Cys-Phe motif) distinct from the XRG lineage [32]. During the engineering process, we succeeded in spectral tuning the dark state by introducing the replacement of two amino acid residues and modulating the d-ring geometry. Notably, one of the replacements was similar to the Tyr/Leu replacement at the same position on the primary sequence as the case for this study (Figure 6). Furthermore, Leu contributes to the spectral blue-shift, whereas Tyr contributes to spectral red-shift, which is also consistent with this study. Interestingly, only AM1_1499g1 possesses the Tyr residue, whereas the other molecules within this lineage possess the Leu residue instead of the Tyr residue (Figure 6B), indicating that AM1_1499g1 specifically acquired the Tyr residue for spectral tuning, which is quite similar to that of AM1_1870g3. Taken together, convergent evolution to acquire the Tyr residue would occur for the spectral red-shift in these two independent lineages. Spectral tuning to introduce corresponding replacements may be possible for the other CBCRs derived from distinct lineages.

The BV-binding molecules are advantageous for the regulation and visualization of molecules within the mammalian cells. Because a longer wavelength of light deeply penetrates into the mammalian tissues, NpF2164g5_BV4_L_962_Y may be a good developmental platform for a better NIR-fluorescent probe. In this study, we succeeded in detecting the red-shifted NIR fluorescence of NpF2164g5_BV4_L_962_Y from the purified protein (Figure 4C). While a single fluorescence peak was observed in the emission fluorescence spectra at the NIR region, multiple fluorescence peaks were observed in the excitation fluorescence spectra; the main peak was detected in the far-red region, and minor ones were detected in the shorter wavelength red region. Because components corresponding to the minor peaks were not clearly observed in the absorption spectra, minor fluorescent components undetectable by the absorption spectrum would be present for this molecule. Similar heterogeneity has also been reported for the phytochromes and other CBCRs [21,33]. Detailed characterization may solve this issue in future experiments.

We succeeded in detecting the red-shifted NIR fluorescence of NpF2164g5_BV4_L_962_Y from living mammalian cells (Figure 5), although the fluorescence was dim in comparison with that of the iRFP and background molecule. Because the experimental setup using 640 nm excitation light for mammalian cell imaging is disadvantageous for the red-shifted NpF2164g5_BV4_L_962_Y, an improvement in the experimental setup should prove effective. Furthermore, low fluorescence quantum yield should be greatly improved for practical use as a NIR-fluorescent probe. Random mutagenesis is one of the powerful methodologies to obtain brighter molecules [26,27], which we would like to address in the near future.

In this study, we have revealed that pinpoint Leu-to-Tyr replacement is crucial for red-shifting of the BV-binding CBCRs. More comprehensive mutagenesis to modify the D-ring configuration may result in obtaining much more red-shifted molecules, which contributes to further improvement. Furthermore, this study would provide insights into the improvement of the other BV-binding molecules, such as the bacteriophytochrome-based ones.

## 4. Materials and Methods

### 4.1. Bacterial Strains and Growth Media

The *Escherichia coli* strain JM109 (TaKaRa, Shiga, Japan) was used for cloning plasmid DNA. The *E. coli* strain C41 (DE3) (Cosmo Bio, Tokyo, Japan) harboring biliverdin (BV) synthetic system, pKT270, was used for protein expression as previously reported [24,35]. Bacterial cells were grown in Lysogeny Broth (LB) medium containing 20 µg/mL kanamycin with or without 20 µg/mL chloramphenicol.

### 4.2. Bioinformatics

Multiple sequence alignment was constructed using the MEGA7 software [36]. The crystal structures of AnPixJg2_BV4 in the Pfr form (PDB ID: 5ZOH [25]) were utilized to assess the key amino acid residues for spectral tuning. Molecular graphics of AnPixJg2_BV4 in the Pfr form and TePixJg in the Pb form (PDB ID: 4GLQ [34]) were generated using the UCSF Chimera software [37].

### 4.3. Plasmid Construction

Plasmids for protein expression in *E. coli* have been constructed by insertion of gene fragments, AM1_1870g3_BV4 (amino acid positions of the background molecule: 469–676) and NpF2164g5_BV4 (amino acid positions of the background molecule: 862–1033), fused with His-tag sequence on their N-terminal into a pET28a vector (Novagen, Madison, WI, USA) [25]. The plasmids of AM1_1870g3_BV4_Y_605_L and NpF2164g5_BV4_L_962_Y were prepared by site-directed mutagenesis based on each background plasmid. The PrimeSTAR Max Basal Mutagenesis kit reagents (TaKaRa, Shiga, Japan) with appropriate nucleotide primer sets (AM1_1870g3_BV4, forward primer 5′-GAAACCcttCACTACTACCAGATTAAGGCC-3′ and reverse primer 5′-GTAGTGaagGGTTTCCAAATGACAGGG-3′ for the replacement of Tyr_605_ by Leu; NpF2164g5_BV4, forward primer 5′-GAGATTtacGAGAAAATCCAGGCTAAGGCTTAC-3′ and reverse primer 5′-TTTCTCgtaAATCTCTAGGTAGCATTGAACATAACC-3′ for the replacement of Leu_962_ by Tyr; each mutation site is shown by small letters) were used for mutagenesis.

Plasmids for protein expression in mammalian cells were constructed by insertion of the codon-optimized gene fragments, NpF2164g5_BV4 and iRFP, fused with EGFP–flexible linker sequence on their N-terminal into a pcDNA3 vector (Invitrogen, Thermo Fisher Scientific, Waltam, MA, USA) whose cytomegalovirus (CMV) promoter was replaced with the CAG promoter [25]. The plasmid of NpF2164g5_BV4_L_962_Y was prepared by site-directed mutagenesis based on its background plasmid. The PrimeSTAR Max Basal Mutagenesis kit reagents with appropriate nucleotide primer set (forward primer 5′-GAGATTtacGAGAAGATCCAGGCCAAAGCC-3′ and reverse primer 5′-CTTCTCgtaAATCTCCAGATAGCACTGCAC-3′ for the replacement of Leu_962_ by Tyr; the mutation site is shown by small letters) were used for mutagenesis. All plasmid sequences were confirmed by DNA sequencing (Eurofins Genomics).

### 4.4. Protein Expression and Purification

All the proteins were expressed in *E. coli* C41 pKT270, which was cultured in 1 L LB medium at 37 °C until the optical density at 600 nm was 0.4–0.8. The cells were cultured overnight at 18 °C after isopropyl β-d-1-thiogalactopyranoside (IPTG) addition (0.1 mM for AM1_1870g3 variants and 1.0 mM for NpF2164g5 variants). The cells were collected by centrifugation at 5000× *g* for 15 min and then frozen at −80 °C. The cells were suspended in a lysis buffer (20 mM HEPES-NaOH pH 7.5, 0.1 M NaCl, and 10% (*w*/*v*) glycerol) and disrupted by Emulsiflex C5 high-pressure homogenizer at 12,000 psi (Avestin, Inc., Ottawa, Canada, ON, Canada)). The mixtures were centrifuged at 165,000× *g* for 30 min to separate the pellets and the supernatants. The collected solutions were filtered through a 0.2 µm cellulose acetate membrane and loaded onto a nickel-affinity His-trap column (GE Healthcare, Piscataway, NJ, USA) using an ÄKTAprime plus (GE Healthcare, Piscataway, NJ, USA). His-tagged proteins were purified using the lysis buffer containing 100 to 400 mM imidazole with a linear gradient system (1 mL/min, total 15 min) after the column was washed using the lysis buffer containing 100 mM imidazole. The purified proteins were added with EDTA (final concentration, 1 mM), incubated on ice for 1 h, and then dialyzed against the lysis buffer to remove imidazole and EDTA. The Bradford method (Bio-Rad Laboratories, Inc., Hercules, CA, USA) was used to measure the protein concentrations using bovine serum albumin for the standard curve.

### 4.5. Electrophoresis and Zinc-Induced Fluorescence Assay

The purified proteins were diluted in a buffer (60 mM Tris–HCl pH 8.0, 2% (*w*/*v*) sodium dodecyl sulfate (SDS) and 60 mM dithiothreitol (DTT)) for sodium dodecyl sulfate-polyacrylamide gel electrophoresis (SDS-PAGE). The samples were denatured at 95 °C for 3 min and then were electrophoresed at room temperature (r.t., about 20–25 °C) using a 12% (*w*/*v*) acrylamide gel.

To detect the fluorescence of the purified proteins, the electrophoresed gels were soaked in 20 mM zinc acetate at r.t. for 30 min. The gels were exposed by blue light (λ_max_ 470 nm) and green light (λ_max_ 527 nm) using a WSE-5500 VariRays (ATTO, Tokyo, Japan) with a short path filter (passing through < 562 nm) to visualize the fluorescence of the proteins through a long path filter (passing through > 600 nm). The fluorescence bands were imaged using a WSE109 6100 LuminoGraph (ATTO, Tokyo, Japan). After the observation, the gels were stained with Coomassie brilliant blue R-250 (CBB).

### 4.6. Spectroscopic Analysis

Ultraviolet and visible absorption spectra of native NpF2164g5_BV4 and BV4_L_962_Y were recorded at r.t. using a UV-2600 spectrophotometer (SHIMADZU, Kyoto, Japan), whereas those of native AM1_1870g3_BV4 and BV4_Y_605_L were recorded at 10 °C to repress dark reversion. An Opto-Spectrum Generator (Hamamatsu Photonics, Inc., Hamamatsu, Japan) was used to generate monochromic near-infrared (NIR) light (700–720 nm) for photoconversion. To monitor the absorption spectra during light irradiation, a light-system was set up, in which the measuring light of the UV-Vis spectrometer vertically crossed the actinic light of the generator.

Fluorescence excitation and fluorescence emission spectra of NpF2164g5_BV4 and BV4_L_962_Y were recorded at r.t. and low temperature (−196 °C, using liquid nitrogen) using a RL-6000 spectrofluorophotometer (SHIMADZU, Kyoto, Japan), in which the excitation light vertically crossed the detecting path of the emission light. Unde rr.t., these fluorescence excitation spectra were monitored by emission wavelength at 710 and 740 nm, respectively, while these fluorescence emission spectra were monitored by excitation wavelength at 640 and 657 nm, respectively. Under the low temperature, these fluorescence excitation spectra were monitored by emission wavelength at 720 and 740 nm, respectively, while these fluorescence emission spectra were monitored by the same excitation light as the r.t. experiments. These fluorescence spectra were corrected by the automatic function of the apparatus.

A Quantaurus-QY (Hamamatsu Photonics, Inc., Hamamatsu, Japan) was used to measure fluorescence quantum yields. The Quantaurus-QY can calculate the absolute value of the fluorescence quantum yield by simultaneously measuring sample absorption and fluorescence emission without known reference standards, as previously described [25]. The excitation wavelength used was 650 nm for NpF2164g5_BV4_L_962_Y bound to BV.

### 4.7. Biochemical and Photochemical Characterization of Cyanobacteriochromes

All the proteins were denatured by 7 M guanidinium chloride (GdmCl)/1% (*v*/*v*) HCl. AM1_1870g3 and NpF2164g5 variants in the dark state were treated in the dark at r.t., whereas AM1_1870g3 variants in the photoproduct state were treated just after turning off the far-red light on ice. The absorption spectra of the denatured proteins were measured at r.t. before and after white light irradiation. To identify the chromophore species and configuration, their absorption spectra were compared with those of the BV-bound AnPixJg2_BV4 [25].

To monitor the photoconversion and dark reversion processes, absorbance at 718 and 708 nm against the far-red light (720 and 710 nm) was measured for AM1_1870g3_BV4 and BV4_Y_605_L, respectively, for 2 min with dark intervals of 5 min at 25 °C. Dark reversion half-lives were calculated from the dark reversion kinetics.

### 4.8. Mammalian Cell Culture and Transfection

HeLa cells were cultured in Dulbecco’s Modified Eagle’s medium (Nissui, Tokyo, Japan) supplemented with 10% fetal bovine serum (SIGMA, St. Louis, MO, USA) and penicillin/streptomycin (Gibco, Thermo Fisher Scientific, Waltam, MA, USA). At 24 h before transfection, HeLa cells were plated onto 35-mm glass-base dishes (IWAKI, AGC Techno glass, Shizuoka, Japan) with a medium containing 25 μM BV (Frontier Scientific, Inc., Logan, UT, USA), which was added from a stock solution. The stock solution of BV was prepared at 25 mM in dimethyl sulfoxide (DMSO, WAKO, Osaka, Japan). Transfection was performed using FuGENE HD Transfection Reagent (Promega, Madison, WI, USA) according to the manufacturer’s instructions.

### 4.9. Confocal Fluorescence Imaging

HeLa cells were transfected using EGFP, EGFP-hiRFP, EGFP-hNpF2164g5_BV4, or EGFP-hNpF2164g5_BV4_L_962_Y. The cells were replaced with a medium containing 25 μM BV 24 h after transfection. The cells were washed and replaced with Opti-MEM (Gibco, Thermo Fisher Scientific, Waltam, MA, USA), 48 h after transfection, and imaged using a Nikon A1 confocal microscope equipped with a Plan-Apochromat 40 × objective. NIR fluorescence of the transfected cells was detected at 663–738 nm upon excitation at 640 nm. Green fluorescence of the transfected cells was detected at 500–550 nm upon excitation at 488 nm.

## Figures and Tables

**Figure 1 ijms-21-06278-f001:**
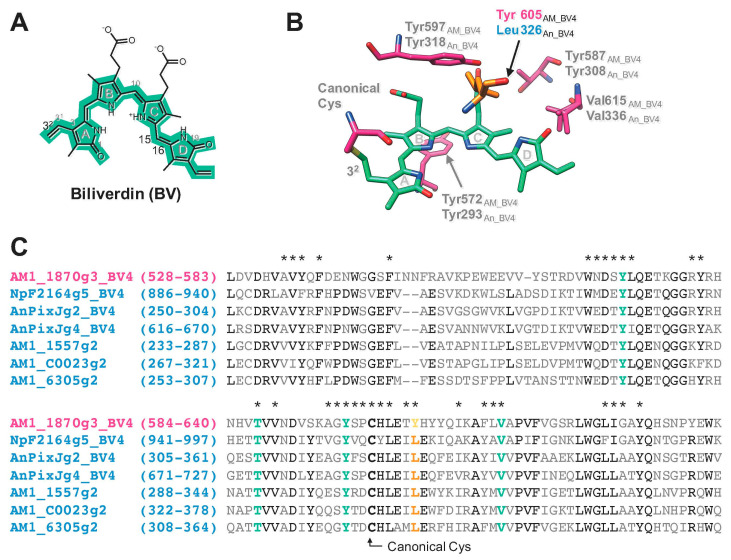
In silico analysis of the XRG CBCR GAF domain possessing BV4 residues. (**A**) Chemical structure of biliverdin (BV). The C3^2^ position on a side chain of BV is covalently bound to a conserved canonical Cys residue within the expanded red/green (XRG) CBCR GAF domains having BV4 residues. *Z*/*E* isomerization of a double bond between the C15 and C16 positions occurs during the photoconversion or dark reversion of their domains. Generally, linear tetrapyrrole pigments (or bilin pigments) incorporated into CBCR GAF domains form *Z*–isomer in dark states (or ground state) and *E*–isomer in the photoproduct states (or excited state). (**B**) The crystal structure of the Pfr form of AnPixJg2_BV4 containing BV (PDB ID: 5ZOH) [25]. Tyr/Leu position is important for spectral tuning (orange), BV4 residues, and canonical Cys (deep pink) with BV (light green) are shown as a stick model. The subscripts, AM_BV4 and An_BV4 on each amino acid number, mean amino acid residues of AM1_1870g3_BV4 and AnPixJg2_BV4, respectively. (**C**) Sequence comparison between AM1_1870g3_BV4 (red-shifted XRG CBCR GAF domain, magenta) and the other XRG CBCR GAF domains having BV4 residues (cyan). Tyr/Leu position, orange; BV4 residues, light green; highly conserved residues, black. Asterisks are shown as amino acid residues within 6 Å of the chromophore.

**Figure 2 ijms-21-06278-f002:**
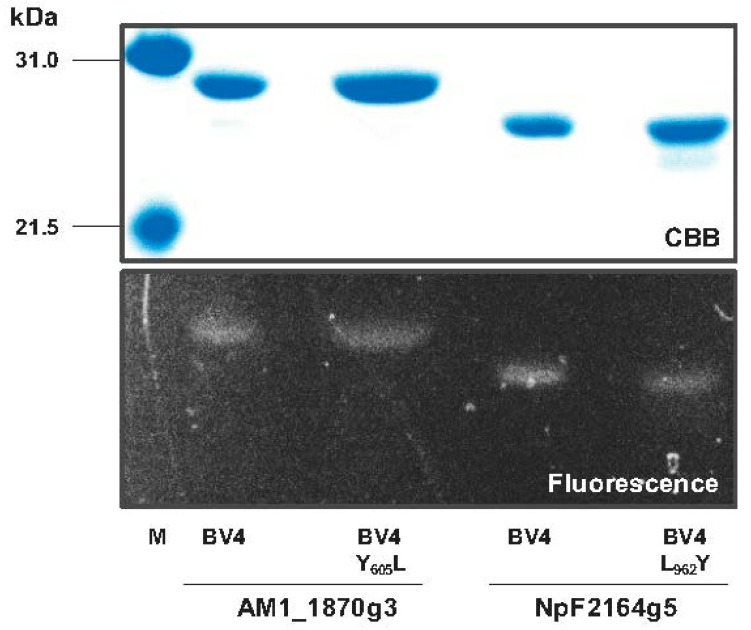
SDS-PAGE analyses of BV-bound AM1_1870g3_BV4, NpF2164g5_BV4, and their variants compared with marker proteins (M) (**upper** panel, CBB stained; **lower** panel, fluorescence imaging).

**Figure 3 ijms-21-06278-f003:**
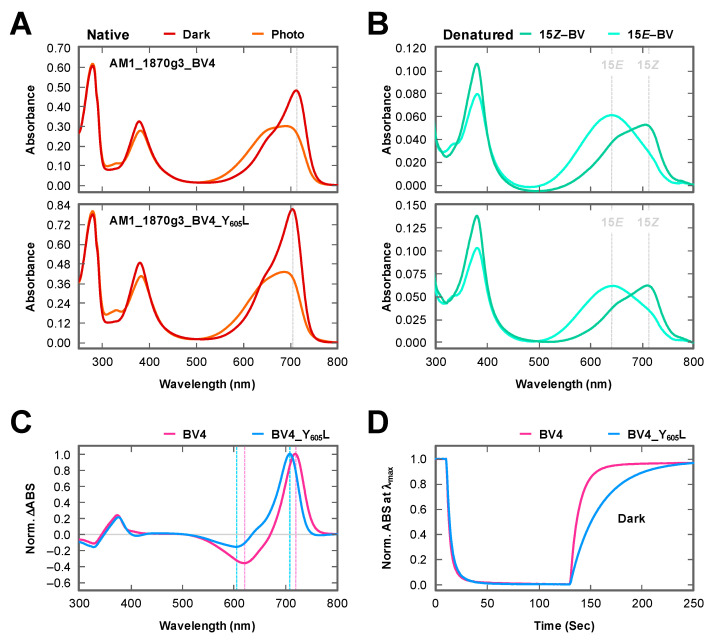
Photochemical properties of BV-bound AM1_1870g3_BV4 and BV4_Y_605_L. (**A**) Absorption spectra of AM1_1870g3_BV4 (**upper** panel) and BV4_Y_605_L (**lower** panel) in dark states (Pfr forms, dark red), and the photoproduct states (orange) under native conditions. These native spectra were recorded at 10 °C to repress dark reversion. Each peak of the Pfr form was indicated by the dotted line. (**B**) Absorption spectra of AM1_1870g3_BV4 (**upper** panel) and BV4_Y_605_L (**lower** panel) in dark states (15*Z* forms, light green), and the photoproduct states (15*E* forms, light blue) under acid-denatured conditions. The acid-denatured spectra were recorded at r.t. and compared with those of BV-bound AnPixJg2_BV4 [25]. (**C**) Normalized difference spectra (dark state—photoproduct state) of AM1_1870g3_BV4 (magenta) and BV4_Y_605_L (cyan). Positive peaks corresponding to the dark states and negative peaks corresponding to the photoproduct states were indicated by dotted lines (BV4, pink; BV4_Y_605_L, blue). (**D**) Photoconversion and dark reversion cycles of AM1_1870g3_BV4 (magenta) and BV4_Y_605_L (cyan). These processes between far-red light irradiation and dark incubation were monitored at the absorption maxima of each Pfr form (BV4, 718 nm; BV4_ Y_605_L, 708 nm) at 25 °C.

**Figure 4 ijms-21-06278-f004:**
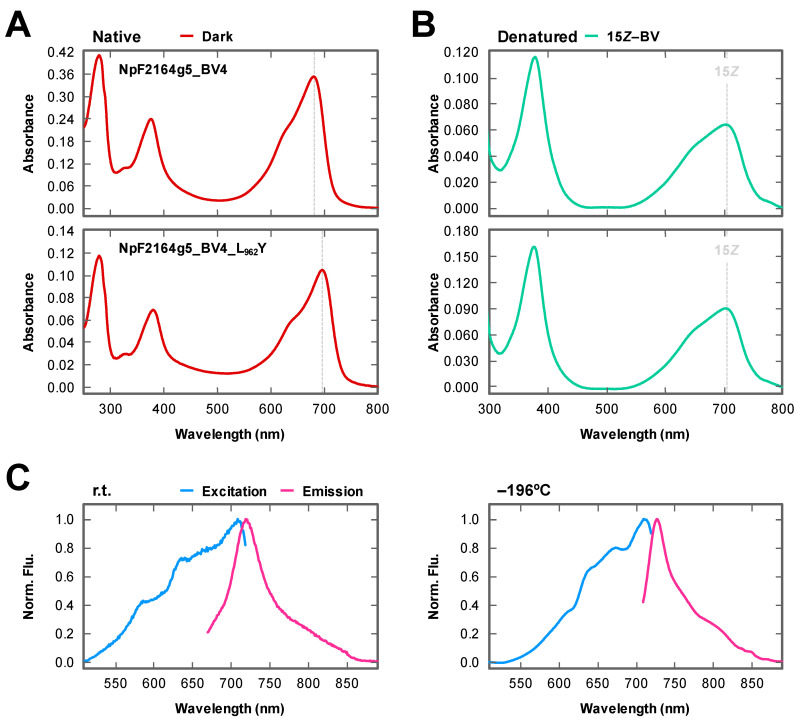
Photochemical properties of BV-bound NpF2164g5_BV4 and BV4_L_962_Y. (**A**) Absorption spectra of NpF2164g5_BV4 (**upper** panel) and BV4_L_962_Y (**lower** panel) in dark states (Pfr forms, dark red) under native conditions. These native spectra were recorded at r.t. Each peak of the Pfr form was indicated by dotted line. (**B**) Absorption spectra of NpF2164g5_BV4 (**upper** panel) and BV4_L_962_Y (**lower** panel) in the dark states (15*Z* forms, light green) under the acid-denatured condition. The acid-denatured spectra were recorded at r.t. and compared with that of BV-bound AnPixJg2_BV4 [25]. (**C**) Normalized fluorescence excitation (cyan) and fluorescence emission spectra (magenta) of NpF2164g5_BV4 (light colors) and BV4_L_962_Y (deep colors). These fluorescence spectra were recorded at r.t. (**left**) and −196 °C (**right**).

**Figure 5 ijms-21-06278-f005:**
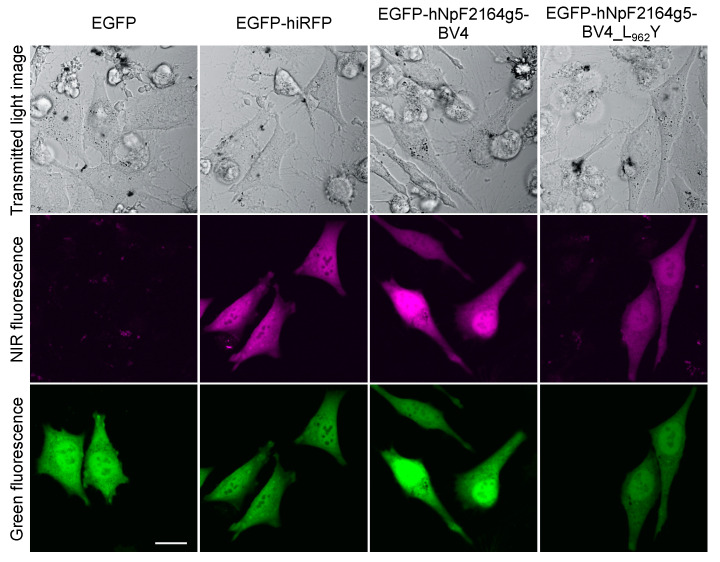
Fluorescence imaging in the HeLa cells using NpF2164g5_BV4_L_962_Y. Transmitted light image (**upper**), NIR fluorescence detection (**middle**), and green fluorescence detection (**lower**) for the HeLa cells expressing EGFP, EGFP-hiRFP, EGFP-hNpF2164g5_BV4, and EGFP-hNpF2164g5_BV4_L_962_Y. The cells were imaged using a confocal microscope, 48 h after transfection. Bar = 20 µm.

**Figure 6 ijms-21-06278-f006:**
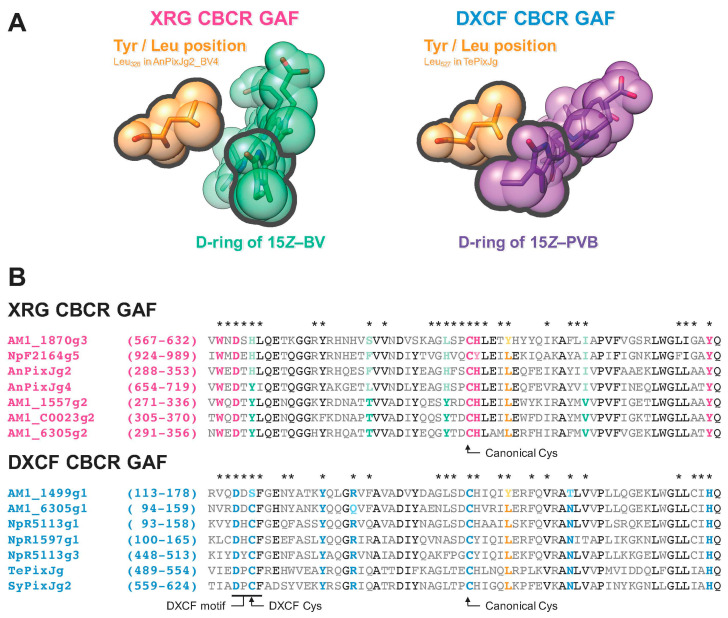
Tyr/Leu position is conserved in various CBCR GAF domains. (**A**) Leu residues (orange) on the Tyr/Leu position of XRG CBCR GAF domain (Leu_326_ in AnPixJg2_BV4 of the Pfr form with 15*Z*–BV (light green), PDB ID: 5ZOH [25]) and of the DXCF CBCR GAF domain (Leu_527_ in TePixJg in the Pb form with 15Z–PVB (phycoviolobilin, red purple), PDB ID: 4GLQ [34]). These structures are shown by the stick and sphere models. (**B**) Sequence alignments of the XRG CBCR GAF (magenta) and DXCF CBCR GAF (cyan) domains. The Tyr/Leu position, orange; key residues for photosensory roles in the XRG (magenta), and the DXCF (cyan) groups; BV4 residues, light green; highly conserved residues, black. Asterisks are shown as amino acid residues within 6 Å of each chromophore.

**Table 1 ijms-21-06278-t001:** Biochemical and photochemical properties of BV-bound AM1_1870g3_BV4 and NpF2164g5_BV4 variants.

	**Absorption Maximum (Dark State) ^a^**		**Dark State–Photoproduct State**		**Dark Reversion**
	**λ_max_ (nm)**	**SAR ^b^**	**Positive (nm)**	**Negative (nm)**	**Half Life (sec) ^c^**
**AM1_1870g3**					
**BV4**	713	0.79	718	619	7.3 ± 0.03
**BV4_Y_605_L**	704	0.85	708	605	20.6 ± 0.11
	**Absorption Maximum (Dark State) ^d^**		**Fluorescence Maximum ^e^**		**Fluorescence**
	**λ_max_ (nm)**	**SAR ^b^**	**Excitation (nm)**	**Emission (nm)**	**Quantum Yield (%) ^f^**
**NpF2164g5**					
**BV4**	680	0.86	696	707	4
**BV4_L_962_Y**	697	0.89	711	728	2

^a^. The absorption maxima in the Pfr forms were calculated from these absorption spectra measured at 10 °C. ^b^. The specific absorbance ratio (SAR) of each variant protein was calculated as the ratio of the Pfr peak absorbance to the protein peak absorbance at 280 nm. ^c^. The half-lives (mean ± standard division, *n* = 3) were calculated from the photoconversion/dark reversion cycles at 25 °C. ^d^. The absorption maxima in the Pfr forms were calculated from these absorption spectra measured at r.t. ^e^. The fluorescence maxima from these fluorescence spectra measured at −196 °C. ^f^. The fluorescence quantum yields were calculated from the ratio of fluorescence emission to light absorption. That of NpF2164g5_BV4 has been reported in a previous study that was shown [25].

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
