# Peer review of "The Cruciality of Single Amino Acid Replacement for the Spectral Tuning of Biliverdin-Binding Cyanobacteriochromes"

_ijms, 2020, doi:10.3390/ijms21176278_

Round 1

Reviewer 1 Report

In this manuscript, Fushimi, et al., have described their work in spectral tuning of cyanobacteriochromes using single amino acid mutations. They have shown successfully demonstrated both “blue shifting” and “red shifting” of biliverdin binding photoreceptors. Furthermore, they have demonstrated the applicability of one of the described proteins, NpF2164g5_BV4_L962Y, for in vivo applications.

Some comments:

The manuscript is too sparse on details. The methods utilized in this study are not described adequately and should be greatly expanded. Perhaps more importantly, given the authors prior publication on the topic (ref. 25) where they reported work showing the adaptation of cyanobacteriochromes to work with mammalian billins, the motivation for this work is unclear, i.e.,  the manuscript does not motivate the study very well. That said, there are other issues that need addressing as listed below.

  1. Please include molecular weight markers for SDS-PAGE gel stained with Coomassie brilliant blue. What was the protein yield in E. Coli?

  2. How were the binding efficiencies calculated? What are the implications of the Y605L mutant having a significantly higher binding efficiency? The authors should discuss the advantages and shortcomings of the mutants.

  3. Is figure 4 composed correctly? The top and bottom panels of Fig. 4 (A) and (B) appear to be identical whereas the text mentions red shift by 17 nm (line 112, page 3).

  4. Figure 4 caption mentions (D). This should be (C). There is only one trace present in this panel but the caption mentions multiple traces.

  5. Line 108 – AM1{…} is mentioned instead of NpF{…}

  6. Please mention measurement temperatures in the figure captions.

  7. The role of AM1{…} mutant is not clear in this manuscript. Blue shift AM1{…} is likely unsuited for in vivo applications. Was the AM1{…} mutant used solely to primarily demonstrate the spectral tuning aspects?

Author Response

Responses to the Reviewer 1

In this manuscript, Fushimi, et al., have described their work in spectral tuning of cyanobacteriochromes using single amino acid mutations. They have shown successfully demonstrated both “blue shifting” and “red shifting” of biliverdin binding photoreceptors. Furthermore, they have demonstrated the applicability of one of the described proteins, NpF2164g5_BV4_L962Y, for in vivo applications.

First of all, thanks for your careful reading and positive evaluations. We consider that we could solve all concerns you raised, which greatly contributed to improve our manuscript.

Some comments:

The manuscript is too sparse on details. The methods utilized in this study are not described adequately and should be greatly expanded. Perhaps more importantly, given the authors prior publication on the topic (ref. 25) where they reported work showing the adaptation of cyanobacteriochromes to work with mammalian billins, the motivation for this work is unclear, i.e., the manuscript does not motivate the study very well. That said, there are other issues that need addressing as listed below.

We are quite sorry not to provide full information about the methods and clear motivation in the introduction. We added detailed explanation about the methods applied in this study (please see the Materials and Methods section). Furthermore, we added several sentences in the introduction to clearly show the problems of the molecules developed in the previous study (Lines 100-101) and to clearly state the motivation (Lines 110-112). In response to these added descriptions, we further added corresponding description in the discussion session (Lines 535-539). We believe that these revisions contribute to large improvement of our manuscript. We fully appreciate your sharp advice.

  1. Please include molecular weight markers for SDS-PAGE gel stained with Coomassie brilliant blue. What was the protein yield in E. coli?

We added the molecular weight marker in Fig. 2. In the case of the previous study (Fushimi et al. 2019 PNAS), the binding efficiency and the protein yield in E. coli were quite important to reveal the molecular mechanism for biliverdin incorporation. Conversely, this study focuses not on biliverdin incorporation but rather on the color tuning. Furthermore, we do not consider that the molecules developed in this study are practically used in E. coli. In these contexts, we did not quantitatively evaluate the protein yield in this study, which is exhausting process. Anyway, judging from E. coli cell colors, the variant molecules are highly expressed, which are comparable to the background molecules.

  1. How were the binding efficiencies calculated? What are the implications of the Y605L mutant having a significantly higher binding efficiency? The authors should discuss the advantages and shortcomings of the mutants.

As mentioned above, we rather focus on color tuning mechanism but not on biliverdin incorporation in this study. Furthermore, the estimation method of the binding efficiency applied in the previous study tend to have large variation. In these contexts, we would like to describe the binding efficiency with qualitative evaluation (Lines 170 & 194). We consider that this variant protein is not for applied science but for basic understanding of the color tuning mechanism. Please see below for details.

  1. Is figure 4 composed correctly? The top and bottom panels of Fig. 4 (A) and (B) appear to be identical whereas the text mentions red shift by 17 nm (line 112, page 3).

The top and bottom panels (Fig. 4A & B) are correct, but quite similar to each other. We revised these panels to emphasize the difference of these spectra to add the dotted lines corresponding to the peak wavelength, which would help readers understand the differences.

  1. Figure 4 caption mentions (D). This should be (C). There is only one trace present in this panel but the caption mentions multiple traces.

Thanks very much for your kind notice. We have missed this point, and now corrected.

  1. Line 108 – AM1{…} is mentioned instead of NpF{…}

Thanks very much for your kind notice. We have missed this point, and now corrected (Line 179).

  1. Please mention measurement temperatures in the figure captions.

We revised the figure captions to add the measurement temperatures (Table 1, Fig. 3, and 4).

  1. The role of AM1{…} mutant is not clear in this manuscript. Blue shift AM1{…} is likely unsuited for in vivo applications. Was the AM1{…} mutant used solely to primarily demonstrate the spectral tuning aspects?

Thanks for your comment. Your comment is definitely right. Blue-shifting is not advantageous for application. The mutagenesis of AM1_1870g3_BV4 was performed to understand the color tuning mechanism. In the revised MS, to clearly state this point, we added a sentence between the AM1_1870g3_BV4 part and the NpF2164g5_BV4 part; To transfer this basic insight into applied science, we focused on the BV-binding molecule, NpF2164g5_BV4, which has been proven to be applicable as a NIR-fluorescent probe without photoconversion, although shorter wavelength absorption peaking at 680 nm should be improved [25] (Lines 174-178).  

Reviewer 2 Report

Comments paper: Fushimi et al 2020, IJMS -899449

The paper describes the role of a specific amino acid in the spectral characteristics of biliverdin binding cyanobacteriochromes. To elucidate this, the authors applied a replacement of a single amino acid near the D-ring of the chromophore. They compare several constructs which contain a Tyr- or a Leu residue. The replacement of the Tyr residue by Leu resulted in a blue shift, while the reverse resulted in a red shift in the absorption and fluorescence spectra. An application in fluorescence imaging of HeLa-cells was presented.

The paper can be considered as an sequel of the previous work of the same group published in PNAS (Fushimi et al, 2019, PNAS, 116 (17), 8301-8309).

Although the paper is well written, there are some major issues that should be addressed to.

The absorption and fluorescence spectra were recorded at room temperature. This results in broad spectra with not very good resolution. Low temperature (-196°C) spectroscopy results in much better resolution of spectra. In this paper, the spectra were not further analyzed by deconvolution techniques or first and second derivatives approaches. This means that determination of  the exact absorption and fluorescence maxima is quasi impossible.

The authors mention in line 85-86 (Results, 2.1) “these residues may affect the π-conjugated system of the chromophore contributing to spectral tuning”. In the discussion, they do not pay further attention to this point, neither to the possible implications of the interaction with the solvent on the spectral tuning which is quite important, especially in fluorescence spectroscopy.

Concerning the confocal microscopy, did the authors check for possible bleaching effects due to the excitation; do they consider to use two-photon excitation?

Some specific comments:

Materials & Methods

Line 301/302: how was the monochromic near-infrared light applied? Was this with a side-illumination system while recording the absorption change?

Line 303-307: are the excitation/emission spectra corrected spectra? Are they taken in “front”-mode, at 45° or under another angle? To reduce ‘stray light’, were specific filters used or not?

Line 334: what is a A1 confocal microscope? Is it the same as in the PNAS-paper (Fushimi et al, 2019)? Did you use the “Spectral unmixing and data analysis methods described in the PNAS-paper? If so, mention this here.

Results

Fig.2: give the details of the fluorescence imaging in the legend: excitation/emission wavelength, filters, or in Materials & Methods or here in the legend of the figure.

Fig.3: The absorption spectra show broad peaks with not well defined maxima in the orange/far-red region (600-750 nm). This means that the spectra are quite complex and consists most probably of different spectroscopic species. How then were the absorption maxima and their shifts determined.

For instance line 90: (purified AM1_1870g3_BV4 Y605L) Pfr form (dark state, λmax 704 nm) to a PO form (photoproduct state, λmax687 nm). Why is in this case Po ( with λmax687 nm) called Po with O meaning “orange” where the wavelength range of orange is between 590-635 nm? The λmax687 nm is in the red wavelength region (635 – 700nm), it can be called Pr form as is the case for phytochromes.

The same remark for the absorption characteristics of the background molecule AM1_1870g3_BV4 in line 93/94.

Line 98: What do you mean exactly with “positive peaks” and “negative peaks”?

Fig.4: excitation – emission spectra: same remark as for the absorption spectra. The fluorescence excitation spectrum is very “rough”, see the remark in M & M, line 303-307 and in general comments. The emission spectrum is most probably also a superposition of several spectroscopic species

Fig 5.: see general remark on the confocal microscopy.

Author Response

Responses to the Reviewer 2

Comments paper: Fushimi et al 2020, IJMS -899449

The paper describes the role of a specific amino acid in the spectral characteristics of biliverdin binding cyanobacteriochromes. To elucidate this, the authors applied a replacement of a single amino acid near the D-ring of the chromophore. They compare several constructs which contain a Tyr- or a Leu residue. The replacement of the Tyr residue by Leu resulted in a blue shift, while the reverse resulted in a red shift in the absorption and fluorescence spectra. An application in fluorescence imaging of HeLa-cells was presented.

The paper can be considered as an sequel of the previous work of the same group published in PNAS (Fushimi et al, 2019, PNAS, 116 (17), 8301-8309).

Although the paper is well written, there are some major issues that should be addressed to.

First of all, thanks for your careful reading and positive evaluations. We consider that we could solve all concerns you raised, which greatly contributed to improve our manuscript.

The absorption and fluorescence spectra were recorded at room temperature. This results in broad spectra with not very good resolution. Low temperature (-196°C) spectroscopy results in much better resolution of spectra. In this paper, the spectra were not further analyzed by deconvolution techniques or first and second derivatives approaches. This means that determination of the exact absorption and fluorescence maxima is quasi impossible.

Thanks for your comments. We agree with your idea especially for the fluorescence spectroscopy, and performed low temperature fluorescence spectral measurements, although we consider that the absorption spectra in our manuscript are acceptable quality to judge the peak wavelength and to evaluate the color tuning issue for in vivo application especially for the dark state (For the photoproduct state, it is difficult to judge the precise peak wavelength because of the incomplete photoconversion. Please see below for details). By the low temperature fluorescence measurement, we succeeded in obtaining high resolution fluorescence spectra, suggesting presence of heterogeneity. We added description about this result in the results and discussion sections (Lines 185-191, 570-577).

We agree with you that deconvolution analysis and first and second derivative approaches are also powerful tool to analyze the photochemical properties. However, we would like to address such detailed chemistry in the future study, because this manuscript is focusing on the spectral tuning and applicability of the red-shifted molecules in vivo.

The authors mention in line 85-86 (Results, 2.1) “these residues may affect the π-conjugated system of the chromophore contributing to spectral tuning”. In the discussion, they do not pay further attention to this point, neither to the possible implications of the interaction with the solvent on the spectral tuning which is quite important, especially in fluorescence spectroscopy.

Thanks for your comments. We fully agree with your idea, and so added description about this issue in the discussion section including the possible effect of the interaction with the solvent (Lines 534-540).

Concerning the confocal microscopy, did the authors check for possible bleaching effects due to the excitation; do they consider to use two-photon excitation?

We performed the confocal microscopy with special care to avoid the bleaching effect, in which we keep minimum irradiation intensity and time to observe the cells. We also checked not so bleached after the observation in comparison with the other area. We do not consider to use the two-photon excitation, because the molecules used in this study absorb already highly red-shifted far-red light with relatively low energy.

Some specific comments:

Materials & Methods

Line 301/302: how was the monochromic near-infrared light applied? Was this with a side-illumination system while recording the absorption change?

Thanks for your comment. You are right. We are sorry for the insufficient explanation about this experiment. We added the experimental details in the Materials and Methods section (Lines 757-759).

Line 303-307: are the excitation/emission spectra corrected spectra? Are they taken in “front”-mode, at 45° or under another angle? To reduce ‘stray light’, were specific filters used or not?

Yes, the spectra are corrected ones by the automatic function of the apparatus applied in this study. This apparatus detected the emission light with 90° angle without specific filters. We added these descriptions in the Materials and Methods section (Lines 709-718).

Line 334: what is an A1 confocal microscope? Is it the same as in the PNAS-paper (Fushimi et al, 2019)? Did you use the “Spectral unmixing and data analysis methods described in the PNAS-paper? If so, mention this here.

Thanks for your comment. We are sorry for the insufficient description. “A1” is the product name produced by Nikon company. To avoid confusion, we revised this sentence as “ Nikon A1 confocal microscope”. We did not perform spectral unmixing and data analysis in this study. This method is specialized to the mouse liver imaging.

Results

Fig.2: give the details of the fluorescence imaging in the legend: excitation/emission wavelength, filters, or in Materials & Methods or here in the legend of the figure.

Thanks for your comment. According to your advice, we added the detailed information in the Materials and Methods (Lines 681-686).

Fig.3: The absorption spectra show broad peaks with not well defined maxima in the orange/far-red region (600-750 nm). This means that the spectra are quite complex and consists most probably of different spectroscopic species. How then were the absorption maxima and their shifts determined.

For instance line 90: (purified AM1_1870g3_BV4 Y605L) Pfr form (dark state, λmax 704 nm) to a PO form (photoproduct state, λmax687 nm). Why is in this case Po ( with λmax687 nm) called Powith O meaning “orange” where the wavelength range of orange is between 590-635 nm? The λmax687 nm is in the red wavelength region (635 – 700nm), it can be called Pr form as is the case for phytochromes.

The same remark for the absorption characteristics of the background molecule AM1_1870g3_BV4 in line 93/94.

Thanks for your comments. You are definitely right. We are quite sorry for this inadequate description. We totally revised the corresponding description (Lines 113-129). The broad absorption of the photoproduct is due to incomplete photoconversion, which would be derived from relatively fast dark reversion and low photoconversion quantum yield. Thus, we decided not to mention absorption peaks of the photoproduct states and avoided the naming neither Po nor Pr, only describing as “photoproduct state”. Your comments greatly improved our manuscript in its accuracy.

Line 98: What do you mean exactly with “positive peaks” and “negative peaks”?

Thanks for your comment. The positive peaks are those of the difference spectra, which correspond to the Pfr form, while the negative peaks are those of the difference spectra, which correspond to the photoproduct state. For accurate description, we modified the corresponding sentence (Lines124-126).

Fig.4: excitation – emission spectra: same remark as for the absorption spectra. The fluorescence excitation spectrum is very “rough”, see the remark in M & M, line 303-307 and in general comments. The emission spectrum is most probably also a superposition of several spectroscopic species

Please see comments above. Your comments greatly contributed to improvement of our manuscript. We added discussion description about the several spectroscopic species in the context of heterogeneity (Lines 525-532). We really appreciate your advice.

Fig 5.: see general remark on the confocal microscopy.

Please see comments above. In addition, for precise description, we revised figure explanations: Transmitted -> Transmitted light image, Cy5 -> NIR fluorescence, EGFP -> Green fluorescence. Furthermore, we added an explanation about the experimental process in the figure legend.

Round 2

Reviewer 1 Report

In this revision, the authors have made several changes that improve the manuscript. However, some of these changes are either unclear or incomplete.

Some questions -

  1. Line numbers in the authors’ reply does not appear to match with the manuscript. The only change in the introduction appears to be lines 70 – 72 (“Because longer wavelength … probes for in vivo imaging”). Other line numbers mentioned are also problematic.

  2. The argument regarding binding efficiency is unclear. Is the estimation sufficiently accurate to be reported? The authors argue that there is large variability in these measure. If that is indeed the case, they should provide some error bars and discuss such shortcomings. No such details are found.

  3. Figure 2, panel A – This is an empty box with the text “CBB”.

  4. What concentration of IPTG was used? A rather large range is specified (0.1 mM to 1 mM).

  5. Please specify percentage for PAGE gel either in methods or figure caption.

Overall, there are large gaps in the response that need to be comprehensively addressed.

Author Response

We attached the PDF file with the point-by-point responses and the full manuscript to avoid file transfer errors.

Reviewer 2 Report

Dear authors,

Thank you for the new version and the answers on the comments. But, there is a major problem: in every respons you gave on the comments, you refer to a number of lines. These numbers never correspond to the lines in the new version of the manuscript. For instance, in your answer on the second comment, you refer to lines 185-191, but in these lines, I cannot read the detailed discussion you refer to and where are the lines 570-577? The last line of the manuscript (v2) is 516!! And this is in every of your answers. This makes it very difficult for a reviewer to evaluate the new version of your manuscript.

I marked your answers on the comments in yellow and my comments on the answers in blue.

You really have to improve your manuscript.

Author Response

(The authors gave the same response as above.)

Round 3

Reviewer 1 Report

I thank the authors for kindly providing a redlined version that highlighted the changes clearly. With this revised version, the authors have addressed my concerns. 

Reviewer 2 Report

Dear authors,

Thank you for the explanation concerning the problem of the line numbering and for the supplementary answers. In my opinion, your paper can be accepted for publication.